# Single-Cell Profiling of Coding and Noncoding Genes in Human Dopamine Neuron Differentiation

**DOI:** 10.3390/cells10010137

**Published:** 2021-01-12

**Authors:** Fredrik Nilsson, Petter Storm, Edoardo Sozzi, David Hidalgo Gil, Marcella Birtele, Yogita Sharma, Malin Parmar, Alessandro Fiorenzano

**Affiliations:** Developmental and Regenerative Neurobiology, Wallenberg Neuroscience Center, Lund Stem Cell Centre, Department of Experimental Medical Science, Lund University, 22184 Lund, Sweden; fredrik.nilsson.5372@med.lu.se (F.N.); petter.storm@med.lu.se (P.S.); edoardo.sozzi@med.lu.se (E.S.); da4206hi-s@student.lu.se (D.H.G.); marcella.birtele@med.lu.se (M.B.); yogita.sharma@med.lu.se (Y.S.)

**Keywords:** human pluripotent stem cells, dopamine neuron differentiation, single-cell RNA sequencing

## Abstract

Dopaminergic (DA) neurons derived from human pluripotent stem cells (hPSCs) represent a renewable and available source of cells useful for understanding development, developing disease models, and stem-cell therapies for Parkinson’s disease (PD). To assess the utility of stem cell cultures as an in vitro model system of human DA neurogenesis, we performed high-throughput transcriptional profiling of ~20,000 ventral midbrain (VM)-patterned stem cells at different stages of maturation using droplet-based single-cell RNA sequencing (scRNAseq). Using this dataset, we defined the cellular composition of human VM cultures at different timepoints and found high purity DA progenitor formation at an early stage of differentiation. DA neurons sharing similar molecular identities to those found in authentic DA neurons derived from human fetal VM were the major cell type after two months in culture. We also developed a bioinformatic pipeline that provided a comprehensive long noncoding RNA landscape based on temporal and cell-type specificity, which may contribute to unraveling the intricate regulatory network of coding and noncoding genes in DA neuron differentiation. Our findings serve as a valuable resource to elucidate the molecular steps of development, maturation, and function of human DA neurons, and to identify novel candidate coding and noncoding genes driving specification of progenitors into functionally mature DA neurons.

## 1. Introduction

Dopaminergic (DA) neurons in the ventral midbrain (VM) are essential for controlling key functions such as control of voluntary movement, reward processing, and working memory. Dysfunction of DA neurons in the ventral tegmental area (VTA), which projects into corticolimbic structures, is associated with the development of neuropsychiatric disorders, drug addiction, and depression while degeneration of DA neurons in the substantia nigra compacta (SNc) is the main pathology in Parkinson’s disease (PD). Over the last three decades, scientific endeavors have focused on designing a novel cell-based therapy for PD via replacement of lost cells with new healthy DA neurons [1]. Major efforts have thus focused on generating ventral midbrain DA neurons from human pluripotent stem cells (hPSCs) for use in disease models and diagnostics [2] as well as in cell-based therapies for PD [3,4]. It is now possible to obtain mature and functional DA neurons from hPSCs for disease modeling and therapeutics [5,6,7], and these 2D culture systems could potentially also be used as stem cell models of human VM DA neurogenesis.

We previously reported a standardized differentiation approach based on floor-plate transition with the addition of dual-SMAD inhibition (Noggin and SB431542) combined with i) sonic hedgehog (SHH) to induce differentiation into ventral neural fates and ii) glycogen synthase kinase 3 inhibitor (GSK3i) to progressively pattern hPSCs toward a caudal VM progenitor phenotype [8]. Our protocol also included the timed delivery of fibroblast growth factor 8b (FGF8b) to drive a more fine-tuned control of rostro-caudal patterning of VM progenitors, which subsequently gave rise to mature DA neurons [9,10].

In this study, we assessed the potential of long-term in vitro differentiation as a stem-cell model of human DA neuron specification and maturation. We used 10× single-cell RNA sequencing (scRNAseq) to perform high-throughput transcriptional profiling of a large number of hPSCs (19,841) at different developmental stages of VM patterning and differentiation. To better dissect the DA developmental program, we examined the cellular composition of VM-patterned stem-cell cultures at the progenitor state, and after one and two months, when functionally mature DA neurons have been formed. Based on single-cell data obtained at different timepoints of DAgenesis, we reconstructed developmental trajectories able to deduce the origin and timing of cell type appearance during VM differentiation. The advent of single-cell resolution technologies has brought unprecedented insights into the complexity and diversity of cell types during brain development [11,12]. However, most scRNAseq studies of neural differentiation currently focus on coding genes, failing to consider the vast amount of noncoding RNA. Long noncoding RNAs (lncRNAs) are a class of noncoding transcripts longer than 200 nucleotides that have emerged as key transcriptional and posttranscriptional regulators acting at multiple levels of gene expression [13,14,15,16]. We therefore developed a bioinformatic pipeline for profiling the expression of lncRNA and obtained a single-cell landscape of these noncoding genes during DA differentiation in the 10× Genomics dataset by mapping their temporal and cell-type-specific expression.

Together, our findings constitute a valuable resource that may help identify both candidate lncRNAs and coding genes involved in regulatory mechanisms during DA neurogenesis. This single-cell dataset will serve as a powerful tool to elucidate human DA neuron development, maturation, and function, and will contribute to establishing more refined DA differentiation protocols.

## 2. Materials and Methods

### 2.1. hPSC Culture and 2D Differentiation

Undifferentiated RC17 (Roslin Cells, #hPSCreg RCe021-A) were maintained on 0.5 μg/cm^2^ Lam-521 (BioLamina, #LN-521)-coated plates in iPS Brew 4 medium (Miltenyi, #130-104-368) until the start of differentiation. They were passaged with 0.5mM EDTA (ethylenediaminetetraacetic acid) roughly every 7 days (with 10µM Y-27632 and seeding density 2500 cells/cm^2^) or when becoming confluent. The cells were differentiated into 2D VM-patterned progenitors using our good manufacturing practice (GMP)-grade protocol [9]. On day 0 of the differentiation the cells were detached with EDTA and seeded (10,000 cells/cm^2^) onto Lam-111 (BioLamina, #LN-111)-coated plates (1 μg/cm^2^) in N2 medium with SB431542 (10 μM), Noggin (100 ng/mL), Shh-C24II (300 ng/mL), CHIR99021 (0.9 μM), and Y-27632 (10 μM). Media was changed on days 2, 4, 7. On day 9 the media was changed to either N2 medium with FGF8b (fibroblast growth factor 8b, 100 ng/mL) or solely N2 medium. The cells were detached with accutase on day 11 and replated (800,000 cells/cm^2^) on Lam-111-coated plates (1 μg/cm^2^) in B27 medium with BDNF (brain-derived neurotrophic factor, 20 ng/mL), AA (ascorbic acid, 0.2 mM), and Y-27632 (10 μM), either with or without FGF8b (100 ng/mL). Media was changed on day 14. On day 16 the cells were either taken for analysis or replated for terminal differentiation. For long-term culture the cells were detached with accutase and replated at a lower density (155,000 cells/cm^2^) on Lam-111-coated plates in B27 medium with BDNF (20 ng/mL), AA (0.2 mM), GDNF (glial cell line-derived neurotrophic factor, 10 ng/mL) + db-cAMP (dibutyryl cyclic adenosine monophosphate, 500 μM), DAPT (notch inhibitor, 1 μM), and Y-27632 (10 μM). Media was changed every 2–3 days until the end of the experiment. For the long-term cultures the cells were plated on Lam-111 with double concentration (2 μg/cm^2^) and after day 25 only 50–75% of the media was changed in order to minimize the risk of detachment.

### 2.2. scRNA-Seq Analysis

Cell suspensions were loaded into a 10× Genomics Chromium Single Cell System (10× Genomics) and libraries were generated using version 3 chemistry according to the manufacturer’s instructions. Libraries were sequenced on Illumina NextSeq500 (400 million reads flow cells) using the recommended read length. Sequencing data was first pre-processed through the Cell Ranger pipeline (10× Genomics, Cellranger count v2) with default parameters (expect-cells set to number of cells added to 10× system), aligned to GrCH38 (v3.1.0) and resulting matrix files were used for subsequent bioinformatic analysis. Seurat (version 3.1.1 and R version 3.6.1, R, Vienna, Austria) was utilized for downstream analysis. Batch effects were removed using the Harmony algorithm (1.0), treating individual 10× runs as a batch. Cells with at least 200 detected genes were retained and the data was normalized to transcript copies per 10,000, and log-normalized to reduce sequencing depth variability. For visualization and clustering, manifolds were calculated using UMAP methods (RunUMAP, Seurat) on 20 precomputed principal components. Clusters were identified by calculating a shared-neighbor graph and then defined (FindClusters, Seurat) with a resolution of 0.2. Identification of differentially expressed genes between clusters was carried out using the default Wilcoxon rank sum test (Seurat). For pseudotime and trajectory analysis Slingshot (1.4) was adopted. For comparison with fetal data Seurats LabelTransfer was used with fetal data as reference object projected the PCA structure onto the data.

### 2.3. LncRNA Quantification

Cell populations identified on the basis of protein-coding genes were used to quantify lncRNA from each cell type across three time points. The expression of lncRNAs was analyzed by extracting cell barcodes for all clusters using Seurat function WhichCells and the original .bam files obtained from the Cellranger pipeline were used to subset aligned files for each cluster (subset-bam tool provided by 10×. Each cluster-specific .bam file was then used as input to count LncRNA expression using FeatureCounts (forward strand). The LncRNA annotation file was downloaded from Gencode (release 36, assembly GRCh38). Differential lncRNA expression analysis was performed using DeSeq2 for both cell types (DA neuron and floor-plate progenitors (FP)) and time points (days 16 and 60) as design (p adj < 0.01). This tool creates a general linear model, assuming a negative binomial distribution. LncRNA that overlapped within a 5kb distance to a protein-coding gene were defined as intragenic. Nearby correlation analysis was performed computing Pearson correlation coefficient using change in protein coding gene expression and lncRNA expression. All data visualization was performed using tidyverse (V1.3), ggplot2 (V3.3.0), and genomic ranges (V1.40.0) running on R (V. 4.0).

### 2.4. qRT-PCR

Total RNAs were isolated on day of analysis using the RNeasy Micro Kit (QIAGEN#74004) according to manufacturer instructions. Approximately 1 μg of RNA was reverse transcribed using Maxima First Strand cDNA Synthesis Kit (Thermo Fisher 2#K1642Invitrogen). cDNA was prepared together with SYBR Green Master mix (Roche#04887352001) using the Bravo instrument (Agilent) and analyzed by quantitative PCR on a LightCycler 480 II instrument (Roche) using a 2-step protocol with a 95 °C, 0.5 min denaturation step followed by a 60 °C, 1 min annealing/elongation step for 40 cycles in total. All quantitative RT-PCR (qRT-PCR) samples were run in technical triplicates, analyzed with the ΔΔCt-method, normalized against the two housekeeping genes ACTB (actin-beta) and GAPDH (glyceraldehyde-3-phosphate dehydrogenase) and results are given as a fold change of expression over undifferentiated hPSCs. Details and list of primers are reported in Appendix A.

### 2.5. Electrophysiology

Prior to recording, cells on coverslips were transferred to a recording chamber containing Krebs solution gassed with 95% O_2_ and 5% CO_2_ at RT and exchanged every 20 min during recordings. The standard solution was composed of (in mM): 119 NaCl, 2.5 KCl, 1.3 MgSO_4_, 2.5 CaCl_2_, 25 glucose, and 26 NaHCO_3_. For recordings, a Multiclamp 700B Microelectrode Amplifier (Molecular Devices) was used together with borosilicate glass pipettes (3–7 MOhm) filled with the following intracellular solution (in mM): 122.5 potassium gluconate, 12.5 KCl, 0.2 EGTA (egtazic acid), 10 HEPES (N-2-hydroxyethylpiperazine-N-ethanesulfonic acid), 2 MgATP, 0.3 Na_3_GTP, and 8 NaCl adjusted to pH 7.3 with KOH, as previously described. Data acquisition was performed with pCLAMP 10.2 software (Molecular Devices, San Jose, CA, United States); current was filtered at 0.1 kHz and digitized at 2 kHz. Cells with neuronal morphology and round cell body were selected for recordings. Resting membrane potentials were monitored immediately after breaking-in in current-clamp mode. Thereafter, cells were kept at a membrane potential of −60 mV to −80 mV, and 500 ms currents were injected from −85 pA to +165 pA with 20 pA increments to induce action potentials. For inward sodium and delayed rectifying potassium current measurements, cells were clamped at −70 mV and voltage-depolarizing steps were delivered for 100 ms at 10 mV increments.

### 2.6. Immunocytochemistry

The cells were washed with PBS and fixed in 4% paraformaldehyde solution for 15 min at room temperature (RT) prior to staining. The cells were pre-incubated in a blocking solution containing 0.1 M PBS with potassium (KPBS) +0.1% Triton +5% serum (of secondary antibody host species) for 1–3 h before the primary antibody solution was added.

The cells were incubated with the primary antibodies overnight at 4 °C and the following day they were washed with KPBS before adding the secondary antibody solution containing fluorophore-conjugated antibodies (1:200, Jackson ImmunoResearch Laboratories, West Grove, PA, USA) and DAPI (4′,6-diamidino-2-phenylindole) (1:500). The cells were incubated with the secondary antibodies for 2 h at RT and finally washed with KPBS. Primary antibodies used were: rabbit anti-TH(tyrosine hydroxylase) (1:1000, AB152, Merck Millipore, Burlington, MA, USA), chicken anti-MAP2 (1:10,000, ab5392, Abcam, Cambridge, United Kingdom), mouse anti-Ki67 (1: 500, ab550609, BD Biosciences, San Jose, CA, USA), mouse anti-FOXA2 (1:1000, ab101060, Santa Cruz, Dallas, TX, USA), rabbit anti-LMX1A (1:1000, AB10533, Merck Millipore), goat anti-OTX2 (1:2000, AF1979, R&D Systems, Minneapolis, MN, USA), rabbit anti-TAU (1:2000, A0024, DAKO, Glostrup, Denmark), rabbit anti-GIRK2 (G-protein-regulated inward-rectifier potassium channel 2) (1:500, APC006, Alomone Labs, Jerusalem, Israel), rabbit anti-AADC (aromatic L-amino acid decarboxylase) (1:500, Merck Millipore AB1519), rabbit anti-CALB (calbindin) (1:200, cb38, Swant Inc., Marly, Fribourg, Switzerland), and mouse anti-GFAP (glial fibrillary acidic protein) (1:200 SMI 21, BioLegend San Diego, CA, USA).

### 2.7. Microscopy

Fluorescent images were captured using a Leica DMI6000B widefield microscope with a Leica 12V 100W Lamphouse 11,504,103 BZ: 00 light source. The images were either taken with a 10× objective (HC PL FLUOTAR 10×, NA = 0.30, immersion = DRY) or a 20× objective (HC FLUOTAR L 20x, NA = 0.4, immersion = DRY). The channels and filters used were DAPI (EX=excitation (nm), EM= emissions (nm); EX: 320–400, EM: 430-510), Cy2 (EX: 430–510, EM: 475–575), Cy3 (EX: 485–585, EM: 535-685), and Cy5 (EX: 560–680, EM: 625–775). Image acquisition software was Leica LAS X and images were processed using Adobe Photoshop CC 2020 (Adobe, San Jose, CA, USA). Any adjustments were applied equally across the entire image, and without the loss of any information. All brightfield images were taken with an Olympus CKX53 with a 4x objective (UPlanFL N 4×/0.13 NA, immersion = DRY) and the software used was OLYMPUS cellSens Standard V1.18.

### 2.8. Statistical Analysis

All data are expressed as mean ± standard error of the mean (SEM). A Shapiro–Wilk normality test was used to assess the normality of the distribution and parametric or nonparametric tests were performed accordingly. Statistical analyses were conducted using GraphPad Prism 8.0 (GraphPad, San Diego, CA, USA).

### 2.9. Code Availability

Code for bioinformatics pipeline used in this analysis is available at GitHub https://github.com/davhg96/SC_lncRNA.git.

## 3. Results

### 3.1. hPSCs Are Patterned into Functionally Mature DA Neurons

We used a well-established differentiation protocol [9] that relies on dual-SMAD inhibition for neuralization, combined with exposure to the ventralizing secreted factor SHH and to GSK3i for caudalization in order to efficiently drive hPSC differentiation into DA progenitors and subsequently into mature VM DA neurons for single-cell sequencing (Figure 1A). This protocol results in the formation of a homogeneous VM progenitor population after 16 days, as assessed by immunostaining for FOXA2, LMX1a, and OTX2 (Figure 1B,E and Appendix A). Subsequent differentiation (Figure 1C,D) led to downregulation of the progenitor markers *SOX2* and *SHH1*, and upregulation of the DA transcripts *NURR1* and *TH* (Appendix A). The progressive formation of DA neurons was confirmed by staining for TH/MAP2, TAU/MAP2 and AADC/MAP2 (Figure 1F,G and Appendix A), showing an increase in morphological complexity of cultures over time as DA progenitors exited from cell cycle and differentiated into postmitotic DA neurons. Immunocytochemistry for Ki67 showed that cells were highly proliferative at days 16 and 30, but that proliferation was minimal at day 60 (Figure 1E–G). At this later timepoint, DA neurons had acquired a subtype identity, expressing GIRK2 (Figure 1H) and CALB (Figure 1I), and had reached functional maturity, as confirmed by whole-cell patch-clamp electrophysiological recordings (Figure 1J–L). Patched cells (*n* = 5) exhibited functional properties such as hyperpolarized resting membrane potentials (−47.12 mV) and presence of inward sodium (Na^+^)-outward delayed-rectifier potassium (K^+^) currents (Appendix A). Neuronal function was confirmed by the ability to fire multiple induced-action potentials (APs) upon current injections (*n* = 4/5) (Figure 1J) with majority of the cells (*n* = 4/5) showing rebound APs after brief depolarization (Figure 1K), characteristic of midbrain DA neurons in vitro [17]. Furthermore, some cells (*n* = 3/5) displayed the ability to fire APs without current injection, as shown by spontaneous firing (Figure 1L), indicative of functional neuronal maturation.

### 3.2. scRNAseq Reveals Cell-Type Specificity and Developmental Trajectories During VM Differentiation

We next performed a 10× Genomics droplet-based single-cell time course transcriptomic analysis at days 16, 30, and 60 of differentiation (Figure 1A and Appendix A), and a total of 19,841 cells were retained for analysis following quality control (QC). Uniform manifold approximation and projection (UMAP) and graph-based clustering assigned the majority of cells to either a floor plate or a DA progenitor/neuron identity in the integrated dataset (Figure 2A,B). A few cells with features of vascular leptomeningal cells (VLMCs), a new cell type associated with vasculature in the brain [18], but none with a glia cell signature were detected (Figure 2A,B). These findings were confirmed at protein level by immunocytochemistry (Appendix A). Cell cycle analysis corroborated immunostaining data (Figure 1E–G), with 33%, 6%, and <1% of cells in active cell cycle at day 15, 30, and 60, respectively (Figure 2C,D). A large population of floor-plate cells was also detected, and further analysis of cell cycle showed that the major segregation within this population was due to cycling genes (Figure 2B–D). UMAP distinguished three clusters, which we named FP-1, FP-2, and FP-3 (Figure 2A). We then identified the most highly differentially expressed genes using the Wilcoxon rank sum test for each cell cluster. FP-1 and FP-2 shared key molecular features of radial glial (RG) cells, characterized by expression of *SOX2*, *PLP1*, *EDNRB*, and *SOX9* (Figure 2B and Appendix A). FP-1 differed primarily from FP-2 in that it included cycling cells with a highly proliferative signature (*TOP2A* and *CENPA*), as visualized by UMAP cell cycle score (S.Score and G2M.Score) as well as in expression of the pro-neural gene *SOX2* and the chromatin-associated gene *E2F1*(Figure 2B–D and Appendix A). The FP-3 cluster was instead highly enriched by expression of the morphogens *SHH*, *CORIN*, and *WNt5A* as well as *FOXA2*, *LMX1A*, and *OTX2*, which define the VM floor plate (Figure 2B and Appendix A). Furthermore, the scRNAseq data revealed the absence of pluripotency-associated genes (*NANOG* and *OCT*4) as well as forebrain (*FOXG1* and *SIX6*) and hindbrain (HOXA2 and GBX2) cells, showing efficient VM patterning (Appendix A).

We also identified a large neuronal cluster (8989 cells) with increased expression of *MAP2*, *DCX*, *POU2F2,* and *RBFIX1* (all upregulated with an adjusted *p*-value <10^−30^ compared to FPs) (Figure 2B and Appendix A). This neuronal cluster was further refined into three subclusters, which we called DA-early, DA-1, and DA-2 (Figure 2A and Appendix A). DA-early differed mainly from the DA-1 and DA-2 clusters in its retained expression of the floor-plate markers *LMX1A* and *FOXA2*, but was starting to gain additional expression of the neuronal precursor markers *DCX* and *SYT1*, and the late DA progenitor markers *Nurr1* and *DDC* (Figure 2B and Appendix A). DA-1 and DA-2 subclusters both expressed *TH* (Figure 2B,C). DA-1 (orange in Figure 2A and Appendix A) was characterized by key markers of VM DA neurons such as *TH*, *NR4A2*, *PAX5*, *SLC6A3*, and *EN2* (Figure 2E), while the smaller DA-2 cluster (purple in Figure 2A) displayed lower or no expression of most DA-related genes (Figure 2E). This latter cluster was instead enriched for non-DA genes such as *ANGLT4*, *FEZF1*, *LEFTY2*, *NKX2.1*, and *TBR1* (Figure 2E).

We next compared the molecular identity of the cells in our 2D VM cultures to a newly generated single-cell dataset of human fetal VM from fetuses dissected 6–11 weeks post conception using the Seurats data projection method in order to assess how well the in vitro stem-cell system captures human VM development (Figure 2F). The fetal dataset comprised 23,438 cells, including ~12,400 progenitors and almost 4000 cells classified as DA neurons (Birtele et al) [19]. Overall prediction quality between the 2D VM-patterned stem-cell culture and fetal VM was high, with a median prediction score of 0.67. Accuracy was higher for the FP-1 and the neuronal clusters, and lower for FP-2 (Figure 2G). Focusing on the two DA clusters in 2D culture revealed 43% of the cells in DA-1 and 25% in DA-2 predicted to be in the fetal DA neuron cluster (median prediction score 0.70 and 0.60, respectively) (Figure 2H,I). However, 56% of DA cells in DA-1 and 68% in DA-2 were also predicted to be fetal neuroblasts, suggesting that not all DA neurons fully mature under 2D culture conditions (Figure 2H,I).

The fine-tuned balance of patterning factors is crucial for ensuring homogeneous VM cell commitment. We previously showed that addition of FGF8b to VM progenitors at days 9–16 resulted in more robust and precise DA neuron formation after transplantation in a PD rat model [8,9]. In this study, we compared the cellular composition of terminally differentiated VM cultures patterned with or without FGF8b at single-cell level (Figure 3A). scRNAseq showed that the absence of FGF8b correlated with ectopic expression of rostral meso/diencephalic markers (*BARHL1*, *NKX2.1*, *PAX6*, and *FOXG1*) accompanied by downregulation of caudal VM markers (*PAX5*, *EN1*, and *NR4A2*) (Figure 3B).

Further investigation of the cellular composition of VM culture with and without FGF8b at day 60 showed that in this 2D model, the timed and dosed delivery of FGF8b also resulted in an increased yield of the DA neuron population (Figure 3C,D).

Interestingly, after two months, over twice the number of cells were found in the DA-1 cluster cultured under FGF8b+ conditions than in the FGF8b- culture (45% vs. 20%, respectively) (Figure 3D). In contrast, a larger DA-early and floor-plate clusters were found in the FGF8b- (DA-early, 8%; FP clusters, 47%) than in the FGF8b+ (DA-early, 4%; floor plate, 20%) (Figure 3C,D) cultures, pointing to the fact that a considerable number of cells with immature identities were still present in 2D cultures grown without FG8b. Single-cell analysis (Figure 3E) and immunohistochemistry of early and late DA markers also confirmed that the absence of FGF8b affected the recapitulation of more mature human DA neurons (Figure 3F,G).

Together, these findings underscore the pivotal role of FGF8b in the induction and specification of DA progenitors into functionally mature DA neurons during VM differentiation.

We also assessed the dynamics of DA neuron development in our stem-cell-based model. Time-based expression analysis of progenitor markers including *CORIN*, *SHH*, and *SOX2* decreased from day 16 to day 60, whereas markers for maturing neurons increased over time (Figure 4A). This prompted us to exploit our a single-cell dataset to temporally reconstruct human DA neurogenesis in in vitro 2D culture using Slingshot on integrated data from all timepoints of differentiation (Figure 4B). Slingshot performs lineage inference using a cluster-based minimum spanning tree, constructing simultaneous principal curves for branching paths through the tree. Three tentative lineages were identified, all originating from the floor-plate population (Figure 4B). Specifically, the resulting plots placed the cells along a continuous path, identifying first FP-1 and then FP-2 as the parent lineage, which subsequently underwent gradual transcriptional changes projecting into three different tentative trajectories (Figure 4B,C). Two of the trajectories shared the same four progenitors (FP-1, FP-2, FP-3, and DA-early), but with DA-1 and DA-2 as terminal states, respectively (Figure 4C,D). In the third trajectory, VLMC was the terminal state, with FP-1, FP-2, and FP-3 as progenitors (Figure 4C,D). Overall, these data confirm that key molecular aspects of DA neurogenesis are recapitulated along a temporal axis in 2D VM culture following a precise developmental program, which eventually gives rise to the generation of functionally mature DA neurons. Interestingly, we also found that the VLMC and DA neuron clusters originated from the same VM floor-plate cells and did not diverge until terminally differentiated into DA neurons or VLMCs, respectively.

### 3.3. 10× Genomics Captures a Repertoire of lncRNAs during VM Differentiation

In order to study tissue-specific expression of lncRNAs and their functional importance as critical regulators in key biological processes we developed a pipeline to examine the developmental dynamics and cell-type specificity of lncRNAs during DA neuron differentiation. The pipeline was designed to comprehensively profile differentially expressed lncRNAs at different timepoints or according to cell-type specificity based on the clusters previously defined using coding gene markers. To identify lncRNAs expressed during VM differentiation, we merged the reads from each cluster and performed lncRNA quantification in a strand-specific manner (see Materials and Methods and Figure 5A). By comparing the expression of lncRNAs and mRNAs, we found that the average number of mapped lncRNA genes at each timepoint was significantly lower than that of mRNA genes. However, no significant changes were detected from day 16 to day 60 (Figure 5B). Our pipeline detected 3802 lncRNA genes, including 1961 sense and 1842 antisense (expression base mean ≥1), and distance matrix of lncRNA expression showed a correlation between cell types and differentiation timepoints (Figure 5C,D and Appendix A). Since lncRNAs can exert either repressive or promoting activities on target genes by coordinating protein and RNA interactions both in cis (on neighboring genes) and in trans (on distant loci), we investigated their genomic distribution. Among 3803 identified lncRNA genes, the majority were intragenic (Appendix A). Pearson’s pairwise correlation analysis showed that lncRNAs exhibited a higher change in expression correlation with their nearest protein coding gene (<5 kb, Figure 5E). Specifically, we found a higher percentage of positive correlations between cis pairs. Length distribution showed that the majority of lncRNAs detected in this pipeline ranged from 300 to 2500 base pairs (Figure 5F). This confirms that the pipeline is able to profile lncRNAs during VM differentiation regardless of genomic location, length of transcript, or transcriptional direction.

### 3.4. Single-Cell Analysis Reveals Dynamics and Cell-Type Specificity of LncRNAs During VM Differentiation

scRNAseq based on coding gene analysis distinguished cell types and defined their developmental dynamics during differentiation. We therefore investigated whether the expression of captured lncRNAs correlated to that of molecularly distinct cell clusters identified during DA neuron specification. C (PCA) clustered 456 single cells into three distinct populations, in line with the three different stages of human VM differentiation analyzed, showing that lncRNAs exhibit high stage specificity following VM developmental dynamics (Figure 6A). The MA plot showed significantly differentially expressed lncRNAs (*p* value adjusted < 0.01) between day 16 and day 60 (Figure 6B). As expected, during VM differentiation we found downregulated well-known lncRNAs such as *NR2F2*-*AS1,* and *LHX5*-*AS1* were shown to play regulatory roles in neural stem-cell differentiation (Figure 6B,E and Appendix A) [20,21]. *LINC01833* (also called *SIX3os*), the long noncoding opposite strand transcript of homeodomain factor *Six3,* known to be involved in neuronal differentiation and retinal cell specification, was also downregulated (Figure 6B,E) [22]. Captured downregulated lncNRAs also included annotated transcripts not yet associated with neurodevelopment, such as *LINC01918* and *AL139393.2*, implicated in thyroid and medulloblastoma carcinoma, respectively [23,24], as well as *SMCR2*, also known as *lncSREBF1*, which plays a role in lipid metabolism (Figure 6B,E and Appendix A) [25].

Differentially expressed upregulated lncRNAs in long-term VM culture included the well-known nuclear paraspeckle assembly transcript 1 (*NEAT1*), overexpressed in the substantia nigra, as well as *H19*, *MALAT1*, and *SNHG12*, previously reported to be enriched at early stages of PD pathogenesis (Figure 6B,E and Appendix A) [26,27,28]. We also found upregulated expression of *AC005062.1* and *AL590302.1*, as well as *LINC02019* and the antisense *SPRY4-AS1*, previously associated with tumorigenesis (Appendix A) [29].

We next examined lncRNA expression based on cell-type specificity by combining single-cell data from days 16, 30, and 60. PCA revealed that lncRNAs clearly segregated into DA and floor-plate clusters, and 140 lncRNAs were found significantly differentially expressed in these two molecularly distinct cell types (Figure 6C). Consistent with these findings, we found lncRNAs associated with brain development to be upregulated in the DA cluster, such as the nuclear-localized *MIAT*, also known as *Gomafu*, reported to be expressed in postmitotic neurons (Figure 6C,D) [30]. The imprinted lncRNA *Meg3* [31] (Figure 6E and Appendix A), expressed in the cortex, was also found enriched in DA neurons, similarly to its neighbor gene *DLK1*, which plays a critical role in DA specification. *LINC01111*, identified as an oncosuppressor in pancreatic cancer [32], and the annotated *LHX1*-*DT* and *AC006387.1* transcripts were also significantly enriched in the DA population (Figure 6F and Appendix A). Noteworthy, ~80% of lncRNAs detected in the DA cell type were also found upregulated at day 60 of VM differentiation. In contrast, *OTX2-AS1* was significantly enriched in the floor-plate cluster together with *TP53TG1*, implicated in carcinoma but not as yet associated with a neuronal phenotype [33] (Figure 6D–F and Appendix A). This cluster of immature cells was also characterized by *SNHG18* [34], known to play an oncogenic role in glioma, *LINC0052* [35], which acts as a sponge for miR-608, and the annotated transcripts *AC026124.1*, *AL132780.2*, and *AC026401.3* (Figure 6D–F and Appendix A).

## 4. Discussion

In this study, we used single-cell sequencing to assess the validity and potential use of stem-cell-based differentiation to model human DA neuron development. By transcriptionally profiling almost 20,000 cells at different time points, spanning the transition from progenitor stage to functionally mature DA neurons, we were able to assess the extent to which this culture system recapitulates DA neuron development. Our findings prompt a number of important observations. Firstly, clustering analysis shows that cell types present during stem-cell differentiation are similar to those present during fetal VM development and that no aberrant cell types are found in the cultures. Secondly, the temporal appearance of different cell types corresponds to the order of progression previously observed in human fetal VM [11] and in xenograft models [10]. Thirdly, the molecular profile of the resulting DA neurons closely matches that of fetal VM-derived DA neurons. Taken together, these considerations support the use of stem-cell-based models to study aspects of human brain development.

When using the single-cell dataset to reconstruct lineage specification and DA neurogenesis, we were able to identify three distinct lineages all originating from the same floor-plate population. Two of the trajectories shared exactly the same progenitors (FP-1, FP-2, FP-3, and DA-early), but then diverged into a large DA population (DA-1) and a smaller population of neurons expressing TH but few other DA-associated markers (DA-2). This latter population also shared a much smaller overlap of genes with DA neurons in fetal VM, indicating that some cells in 2D culture underwent incomplete differentiation and/or insufficient patterning toward a VM regional fate or otherwise aberrantly express TH in non-DAergic cells as is commonly observed in in vitro cultures [36,37,38,39]. Interestingly, this lineage analysis also revealed that the VLMC and DA neuron clusters originated from the same VM floor-plate cells and did not diverge until terminal differentiation into DA neurons or VLMCs had occurred.

Rostro-caudal patterning in VM differentiation protocol can be more precisely controlled using timed delivery of FGF8b [9]. By quantifying cellular composition of cells patterened with and without FGF8 after two months in culture, we showed that FGF8b facilitates the conversion of progenitors into functionally mature DA neurons while enabling efficient differentiation toward a caudal VM pattern. These findings also underscore the need to identify and validate more markers able to predict functional maturation in both long-term culture and in vivo after transplantation.

In the past decades, large-scale genome-wide sequencing has revealed the tissue-specificity and functional relevance of lncRNAs as key regulators in neural development, refuting the paradigm of RNA as simply an intermediary between DNA and protein [40,41,42]. In this expanding view of the genomic landscape, very few scRNAseq studies have as yet provided comprehensive human lncRNA repertoires of different healthy human tissues [43,44,45,46]. Although the relatively low expression levels of lncRNAs make analysis at single-cell resolution challenging, a detailed catalog of cell-type-specific lncRNAs expressed during DA differentiation may increase our understanding of their involvement in the VM developmental program and intricate regulatory DA pathways [47,48]. Here, we exploited 3′-end high-throughput sequencing of 10× Genomics technology to capture polyadenylated noncoding genes and developed a bioinformatic pipeline that provided the first comprehensive dataset of lncRNAs showing cell-type specificity by analyzing different stages of DA neuron differentiation. 3′-end sequencing is not able to capture non polyA transcripts and is less efficient than full-length scRNAseq methodologies. Nevertheless, 10× Genomics performs high-throughput microfluidic profiling of a large number of cells. Thus, although many lncRNAs may individually have low expression levels, the extensive number of cells that is possible to sequence with this method also allows lncRNA expression to be analyzed and mapped to distinct clusters defined by the coding gene expresssion.

We found that many lncRNAs are specific to distinct cell types associated with either progenitors or functionally mature DA neurons. Interestingly, we identified new lncRNAs potentially involved in early steps of DA differentiation, such as *TP53TG1, LINC01833,* and *SNHG18*, found specifically enriched in the floor-plate clusters, as well as two antisense transcripts of the *NR2F1* and *OTX2* coding genes, which play a key role in cortical development and VM floor-plate formation, respectively [49]. Our dataset also includes significantly enriched lncRNAs not previously linked to functionally mature DA cells, such as *LINC01111*, *RFPL1S*, and *AL365361.1*. It also confirms the expression of lncRNAs already associated with DA tissue, including *MIAT*, involved in DA signaling via modulation of DA transmission and neurobehavioral phenotypes [50]. Using a microfluidic approach enabled us to analyze a sufficiently large number of cells and more accurately redefine the feature expression map of lncRNAs which until now were thought to be mainly expressed in late stages of VM differentiation. *NEAT1*, for example, known to exert a neuroprotective role in SNc, was also found highly expressed in our floor-plate subclusters, suggesting its potential role in DA commitment [51]. The imprinted lncRNA *Meg3* also displayed a particular cell-type-specific expression pattern robustly expressed in DA clusters, hinting at its involvement as a critical regulator of DA neuron specification, possibly through crosstalk with its coding neighbor gene *DLK1*. In this scenario, integrating single-cell transcriptomics and epigenomics technologies, such as assay for transposase-accessible chromatin using sequencing (ATAC-seq), will bring valuable insights into the complex and intricate regulatory network involving coding and noncoding genes during stem-cell DA differentiation.

Taken together, these findings provide a valuable transcriptomic dataset of coding and noncoding genes in human DA neurogenesis that may lead to the identification of novel cell types and a better definition of molecular diversity in progenitor and mature DA populations.

## Figures and Tables

**Figure 1 cells-10-00137-f001:**
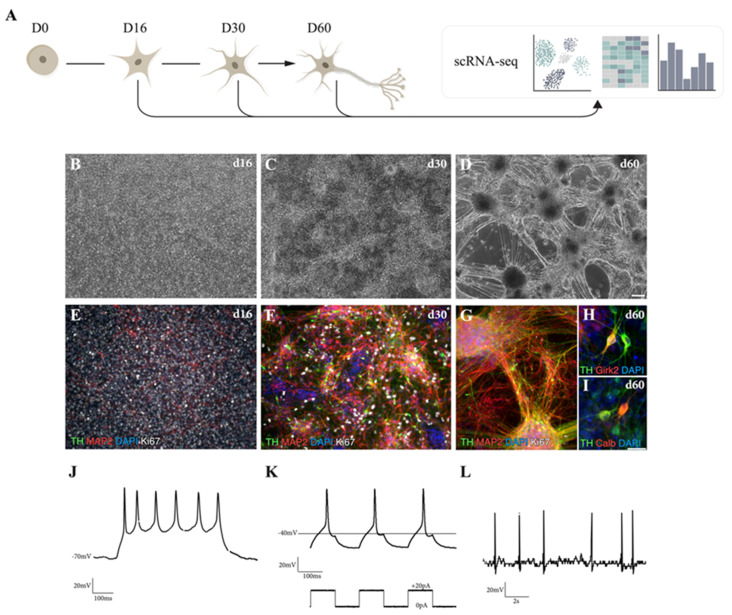
VM-patterned hPSC differentiation generates functionally mature dopaminergic (DA) neurons. (**A**) Schematic overview of the experimental design. (**B**–**D**) Representative bright-field images of ventral midbrain (VM) differentiation cultures at different time points (16, 30, and 60 days). Scale bars, 100 µm. (**E**–**G**) Immunofluorescence staining of tyrosine hydroxylase (TH), MAP2, and Ki67 at days 16, 30, and 60. Scale bars, 100 µm. Nuclei were stained with 4′,6-diamidino-2-phenylindole (DAPI). (**H**) Immunofluorescence staining of DA markers TH/GIRK2. (**I**) TH/calbindin (CALB) at day 60. Scale bars, 25 µm. Nuclei were stained with DAPI. (**J**–**L**) Electrophysiological assessment of DA neuron-rich cultures using patch-clamp analysis. (**J**) Cells analyzed at day 60 displayed induced action potentials. (**K**) Induced action potentials upon brief depolarization. (**L**) Spontaneous firing characteristic of DA neurons.

**Figure 2 cells-10-00137-f002:**
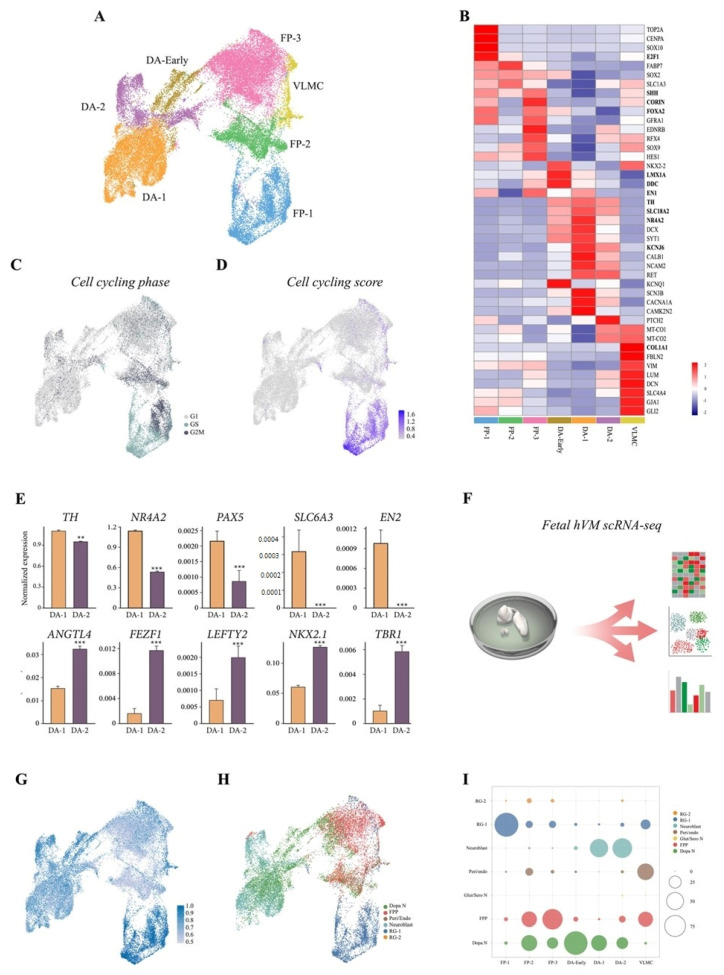
Single-cell RNA sequencing (ScRNAseq) captures distinct cell-type identities during human pluripotent stem cell DA neuron differentiation. (**A**) Uniform manifold approximation and projection (UMAP) plot embeddings of 19,841 cells from VM cultures at day 16, 30, and 60. Cell-type assignments are indicated. (**B**) Heat map showing expression levels of selected genes for each cluster. Values are given as standard deviations relative to average expression across all clusters. (**C**) UMAP embeddings showing the predicted cell cycle phase. (**D**) Cell-cycling score (S.Score + G2M.Score) calculated using Seurat CellCycleScoring function. (**E**) Bar plot of DA-1 and DA-2 clusters showing expression of meso/diencephalic markers. Data represent mean ± SEM, ** *p* < 0.01, *** *p* < 0.001. (**F**) Schematic overview of the experimental design. (**G**) UMAP embeddings of prediction score for human fetal VM cells from three separate fetuses (6, 8, and 11 weeks post-conception) vs. hPSC VM-derived culture cell types. (**H**) Predicted cell types using fetal derived cell types as reference. (**I**) Relative overlapping quantification (% cells overlapping) of human fetal midbrain vs. hPSC VM culture datasets.

**Figure 3 cells-10-00137-f003:**
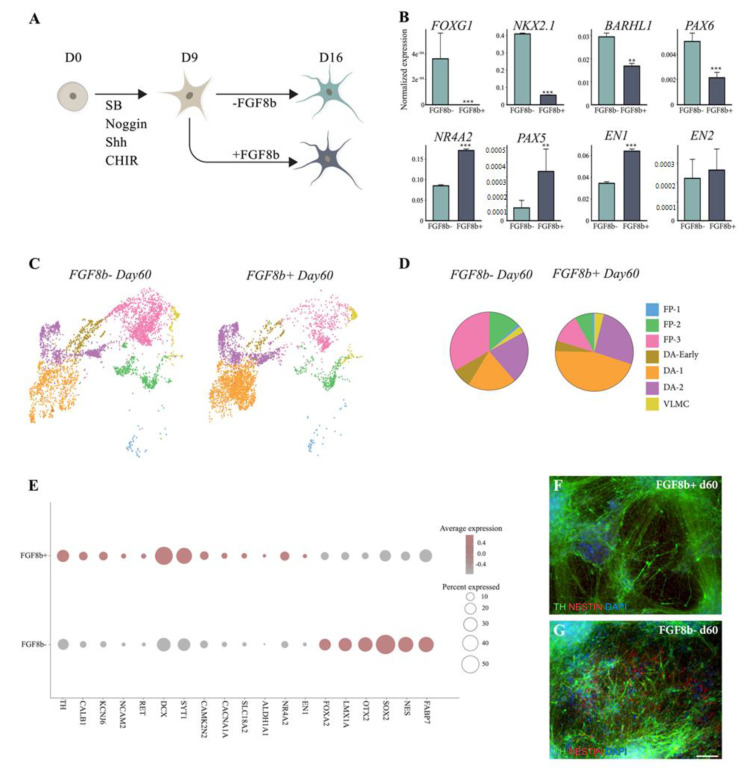
Fibroblast growth factor 8 (FGF8b) facilitates progenitor transition into mature DA neurons. (**A**) Schematic overview of the experimental design. (**B**) Bar plots showing expression of rostral and caudal mesencephalic markers with and without FGF8b treatment at 2 months from single-cell dataset. Data represent mean ± SEM, ** *p* < 0.01, *** *p* < 0.001. (**C**) UMAP embeddings showing cells from 60-day VM culture with and without FGF8b colored by cell types. (**D**) Pie chart showing cell-type proportion with and without FGF8b at 2 months. (**E**) Dot plot showing percent of cells expressing early and late DA markers in VM culture with and without FGF8b at 2 months. (**F**) Immunofluorescence staining of TH/NESTIN with and (**G**) without FGF8b treatment at 2 months VM culture. Scale bars, 100 µm. Nuclei were stained with DAPI.

**Figure 4 cells-10-00137-f004:**
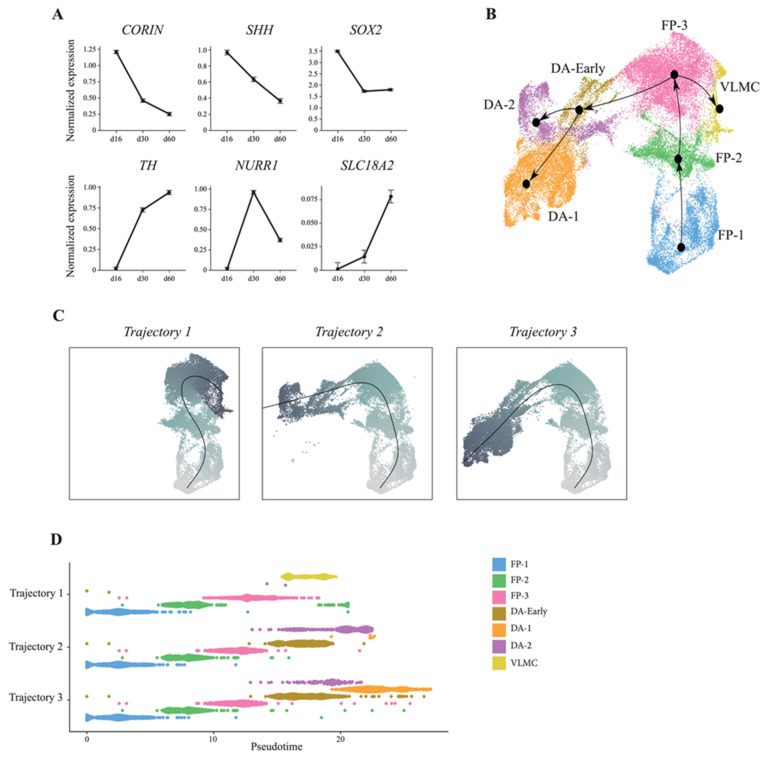
ScRNAseq reconstructs developmental trajectories during VM-patterned hPSC differentiation. (**A**) Temporal expression pattern of early and late DA markers during hPSC VM differentiation (16–60 days). (**B**) Slingshot-based pseudotime trajectories calculated from UMAP embeddings showing combined developmental trajectories of all identified cell types and, (**C**) individual trajectory curves colored by pseudotime. (**D**), Pseudotime inference reconstruction plot showing the appearance and temporal progression of cell-types along a pseudotemporal axis.

**Figure 5 cells-10-00137-f005:**
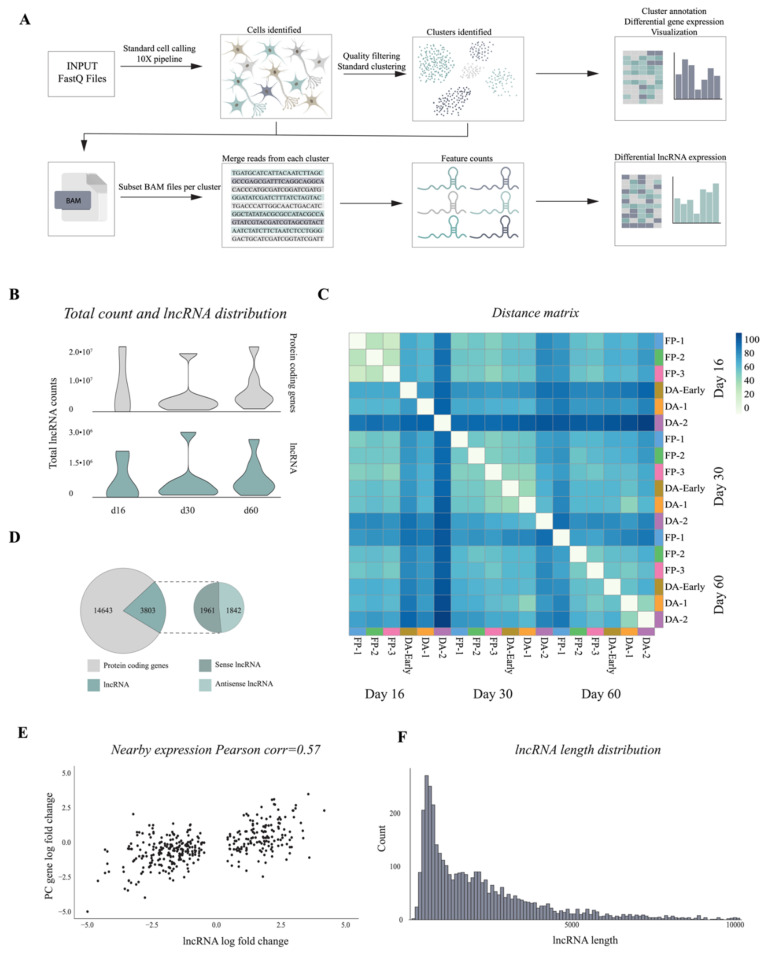
Bioinformatic pipeline detects lncRNAs from 10x Genomics single cell dataset (**A**) Schematic overview of pipeline designed for lncRNA quantification. (**B**) Count distribution of protein-coding genes and lncRNAs from each sample at three time points of hPSC DA neuron differentiation (day 16, 30, 60). (**C**) Heatmap of Euclidean distance across different cell types and time points (log2 normalized counts). (**D**) Number and distribution of protein-coding genes and lncRNAs expressed (baseMean >1), strand (sense “+”, antisense “−”). (**E**) Pearson correlation of change in expression of protein-coding genes and lncRNAs (±5Kb, p adj < 0.01). The x-axis shows the log fold change of the lncRNA and the y-axis the log fold change of the overlapping protein-coding gene. (**F**) Length distribution of expressed lncRNAs (x-axis).

**Figure 6 cells-10-00137-f006:**
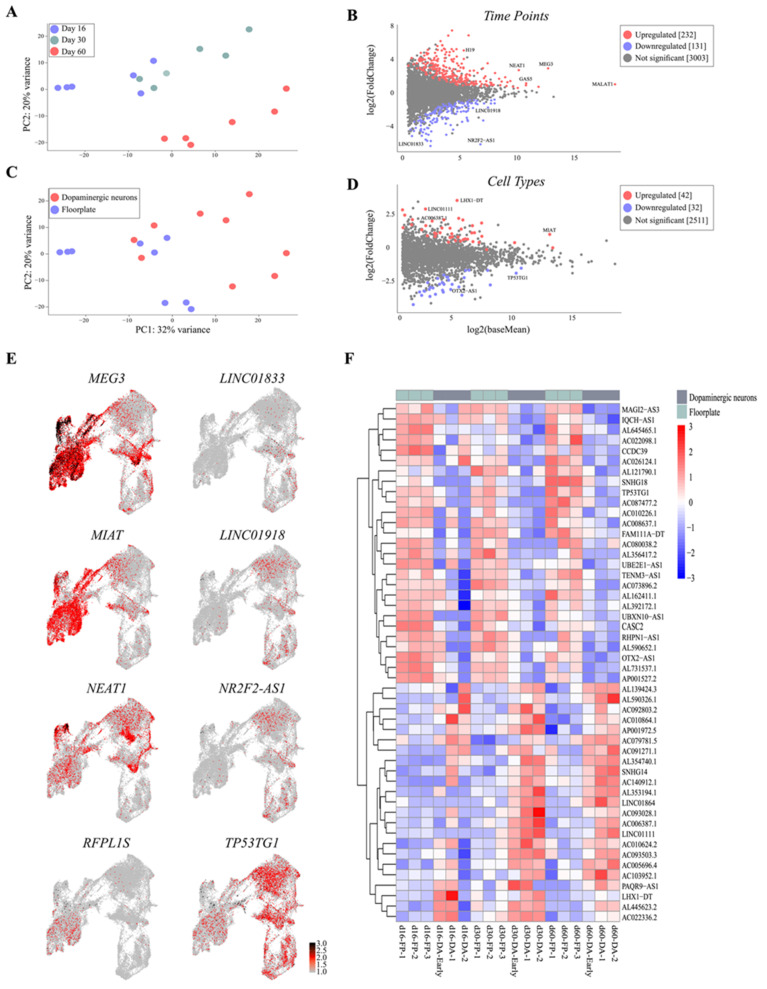
Catalogue of lncRNA time and cell-type specificity. (**A**) Principal component analysis (PCA) plot of lncRNA expression across timepoints in hPSC DA neuron differentiation. (**B**) MA plot of lncRNAs showing their log2 fold change and log2 baseMean expression across time points (*p* adj < 0.01). (**C**) PCA plot of lncRNA expression across cell types in hPSC DA neuron differentiation. (**D**) MA plot of lncRNAs showing their log2 fold change and log2 baseMean expression across cell type (*p* adj < 0.01). (**E**) Expression map of candidate lncRNAs projected on UMAP plot. (**F**) Expression heatmap of the top 50 most differentially expressed lncRNAs across cell types (log2 vst, *p* adj < 0.01).

## Data Availability

The data presented in this study are available in Appendix A.

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
