# Peer review of "Single-Cell Profiling of Coding and Noncoding Genes in Human Dopamine Neuron Differentiation"

_cells, 2021, doi:10.3390/cells10010137_

Round 1

Reviewer 1 Report

The manuscript by Nilsson and colleagues reports a transcriptional single cell RNA sequencing profiling of ventral mesencephalic patterned stem-cells at progressive stages of neuronal maturation. The authors have defined the cellular composition of cultures at different timepoints, revealing (i) homogeneous DA progenitor formation at an early stage of differentiation and (ii) after two months of maturation, dopaminergic neurons with molecular identities resembling those of authentic dopaminergic neurons derived from human ventral mesencephalic fetal tissue.   The authors also set up a novel bioinformatic pipeline useful to generate a precise long noncoding RNA expression network which may help to elucidate to  the complex regulatory RNA-based network established during dopaminergic neuron differentiation. 

Overall the data are clean, the provided information are very interesting and well documented, and the paper clearly written. The results are well presented and in-depth discussed. Conclusions are coherent with the results obtained. 

Author Response

We thank the Reviewer for their generally positive feedbacks and kind comments on the work presented.

Reviewer 2 Report

The article brings data on DA neurons from single cells obtained at different timepoints of DA genesis together with the development of a bioinformatic pipeline for profiling the expression of lncRNA. The article is very well written and very sound, it can be almost accepted as it is. Only, it would be beneficial if all methods are described in greater detail so that they are fully reproducible. Otherwise, it is a great study and it will certainly attract large readers' attention.

For example (but also in other methods details are needed):

qRT-PCR: provide more details, so that it is fully reproducible, also what was used as a standard / which genes?

Microscopy: objective, NA, filters used, light source

Author Response

We thank the Reviewer for their careful reading of our manuscript and constructive suggestions, which have helped us improve the quality of the paper.

Please find below a point-by-point response to all comments.

Based on the Reviewer’s suggestion, we have now improved the background and included relative references in the Introduction (page 1, lines 79-81 and page 2, lines 106-112)

We have now provided a more detailed description of the following Method sections.

hPSC culture and 2D differentiation: page 2, lines 122-144

LncRNA quantification: page 3, lines 184-187

qRT-PCR: page 3, lines 194-204

 Microscopy: page 4, lines 258-267